# Edge States and Strain-Driven Topological Phase Transitions in Quantum Dots in Topological Insulators

**DOI:** 10.3390/nano12234283

**Published:** 2022-12-01

**Authors:** Benjamin Puzantian, Yasser Saleem, Marek Korkusinski, Pawel Hawrylak

**Affiliations:** 1Department of Physics, University of Ottawa, Ottawa, ON K1N 6N5, Canada; 2Quantum Theory Group, Security and Disruptive Technologies, National Research Council of Canada, Ottawa, ON K1A 0R6, Canada

**Keywords:** quantum dots, topological insulators, edge states, quantum strain sensors, topological phase transition

## Abstract

We present here a theory of the electronic properties of quasi two-dimensional quantum dots made of topological insulators. The topological insulator is described by either eight band $\vec{k}\cdot \vec{p}$ Hamiltonian or by a four-band $\vec{k}\cdot \vec{p}$ Bernevig–Hughes–Zhang (BHZ) Hamiltonian. The trivial versus topological properties of the BHZ Hamiltonian are characterized by the different topologies that arise when mapping the in-plane wavevectors through the BHZ Hamiltonian onto a Bloch sphere. In the topologically nontrivial case, edge states are formed in the disc and square geometries of the quantum dot. We account for the effects of compressive strain in topological insulator quantum dots by means of the Bir–Pikus Hamiltonian. Tuning strain allows topological phase transitions between topological and trivial phases, which results in the vanishing of edge states from the energy gap. This may enable the design of a quantum strain sensor based on strain-driven transitions in HgTe topological insulator square quantum dots.

## 1. Introduction

There is currently significant interest in both topological insulators (TIs) [1,2,3,4,5,6,7,8,9,10,11,12] and in semiconductor quantum dots [13,14,15,16,17,18,19,20,21,22,23]. A TI is a semiconductor with an insulated bulk and an energy gap in which the gapless helical states, localized at the edge of the material, were predicted to exist [1,2,3,5,6,7,8]. The interest in TIs was stimulated further by the experimental demonstration of edge states and the spin quantum Hall effect in HgTe/CdTe quantum wells with an inverted band structure [3] and in many other materials [5,6,7,8,9,10,11,12].

Simultaneously, the interest in semiconductor quantum dots is driven by potential applications in lasers, transistors, and single and entangled photon sources as building blocks in quantum technologies [17,18,19,20,21,22,23]. Moreover, interest in HgTe-based nanocrystals and nanoplatelets [24,25] is motivated by their potential application as far-infrared (FIR) detectors [26,27]. In normal semiconductors, edge states, being detrimental to the performance of the devices, are commonly passivated. This improves the performance of the device. By contrast, edge states in topologically nontrivial insulators are robust and give rise to novel physics. For example, one can envisage edge states as realizations of one-dimensional (1D) strongly interacting systems and, as will be discussed in this paper, sites in one-dimensional Hubbard models [28,29]. Edge states also arise naturally in the study of quantum dots in two-dimensional (2D) materials such as graphene [16].

The theory of interface between topological and normal HgTe insulators was developed by, e.g., Volkov and Pankratov [1], Zhang et al. [2], and Fu and Kane [30]. Zhang et al. [2] used $\vec{k}\cdot \vec{p}$ theory to derive the four-band effective 2D Bernevig–Hughes–Zhang (BHZ) Hamiltonian. The BHZ Hamiltonian is one of the simplest Hamiltonians that describes the quantum spin Hall effect and edge states at the interface between inverted band HgTe quantum wells and normal insulators. The BHZ model has also been used to describe different quantum dot structures. Such examples include work conducted by Chang et al. [31,32,33] and Zhu et al. [34], who both numerically analyzed helical edge states in cylindrical quantum dots. Later on, Zhu et al. [35] numerically studied the effects of tensile strain on rectangular quantum dots along the horizontal axis, concluding that irrespective of the deformations, edge states are robust and the energy gap increases. In our earlier work, using an eight-band 3D $\vec{k}\cdot \vec{p}$ model, it was shown that edge states appeared as a function of the thickness of the quantum dot and/or applied strain [36].

Tight binding models [37] have also been used to understand the electronic and optical properties of HgTe TI quantum dots, by, e.g., Peeters et al. [38] and Delerue et al. [39]. Some of these tight-binding models [38,39] have been used to investigate the edge states in HgTe TIs [38] and optical absorption in HgTe quantum rings [37]. Such approaches are, however, numerical in nature and do not allow for simple understanding of the physics of TI quantum dots.

In this paper, we contribute to the theory of the electronic properties of quasi 2D quantum dots made of TIs described by a four-band BHZ model and compare our findings to results obtained by the eight-band $\vec{k}\cdot \vec{p}$ method [36]. The trivial versus topological properties of the bulk BHZ Hamiltonian are established by examining a topology of a mapping of a plane of wavevectors $k_{x},k_{y}$ onto a Bloch sphere. We show that one set of material parameters defines the topologically nontrivial case, in which topologically protected edge states are found, and another set of parameters defines a topologically different map corresponding to a trivial insulator without edge states. Due to the simplicity of the BHZ model, we are able to explicitly relate the emergence of edge states to the material parameters and to the topology of the map. The edge states are characterised by decay length into the center of the quantum dot and by a distance from the physical edge. Analytical results are obtained and successfully compared with numerical results from the eight band $\vec{k}\cdot \vec{p}$ [36].

In this work, we add the Bir–Pikus Hamiltonian [36] to the BHZ Hamiltonian to account for strain, and study the effects of strain on the electronic properties of quantum dots. For the square quantum dot, edge states were controlled by tuning the amount of compressive strain in the quantum dot along both axes. By tuning the strain past a critical value, a topological phase transition occurred, leaving behind a quantum dot made of trivial insulator without edge states in agreement with previous work on quantum discs using the eight-band $\vec{k}\cdot \vec{p}$ model. When our quantum dot is connected to the leads, the conductivity strongly depends on the presence or absence of edge states induced by strain. Hence, strain in the quantum dot can be detected through conductance measurements in this nano strain sensor [36].

This paper is organized as follows: Section 1 starts with an introduction to TI quantum dots. Section 2 describes the BHZ model, its topological properties, effect of strain and energy levels and wavefunctions of two different HgTe TI nanostructures: quantum discs, and unstrained and strained quantum squares. Section 3 discusses numerical and analytical results for energy spectrum, strain induced phase transition and numerical and analytical results for edge states. Finally, in Section 4, we present our conclusions and future directions.

## 2. Models

An example of a quasi-2D TI is a quantum well made of HgTe, embedded in a higher bandgap normal insulator material such as CdTe [2,36]. This structure is described by the effective quasi-2D BHZ $\vec{k}\cdot \vec{p}$ Hamiltonian, in which the wide bandgap insulator is replaced by a vacuum with an infinite energy gap. This avoids the uncertainties in $\vec{k}\cdot \vec{p}$ treatment of material interfaces [40,41] but it implies that the wavefunction of an electron is zero at the physical edge of the nanostructure; hence, we need to determine the position of edge states away from the interface. In this paper, we study finite nanostructures of quasi-2D HgTe quantum dots as illustrated in Figure 1 and described by the BHZ Hamiltonian. The results for the disc will be compared with eight band $\vec{k}\cdot \vec{p}$ approach [36] and then, with BHZ approach validated, the BHZ model will be applied to the square quantum dot.

In the $\vec{k}\cdot \vec{p}$ BHZ theory, the wavefunction of the spin-up electron in a quasi-2D layer of HgTe is written as a linear combination of electron |e↑〉 and heavy hole |hh↑〉 states: $|\varphi_{k}↑\rangle =A_{k}e^{i\vec{k}\cdot \vec{r}}|e↑\rangle +B_{k}e^{i\vec{k}\cdot \vec{r}}|hh↑\rangle$ with a similar expansion for the spin-down electron.

The effective four-band Hamiltonian is given by the spin block diagonal BHZ Hamiltonian:(1)$$H_{BHZ}=\left(\begin{array}{cc} H_{↑} & 0\\ 0 & H_{↓} \end{array}\right).$$

The spin-up Hamiltonian $H_{↑}$, acting on spinors $\left(A_{k},B_{k}\right)$, is given by
(2)$$H_{↑}=\left[\begin{array}{cc} \frac{\Delta (k_{x},k_{y})}{2} & v_{f}\left(k_{x}-ik_{y}\right)\\ v_{f}\left(k_{x}+ik_{y}\right) & -\frac{\Delta (k_{x},k_{y})}{2} \end{array}\right],$$
where $v_{f}$ is the Fermi velocity, $\Delta \left(k_{x},k_{y}\right)=2\left(M+B\left(k_{x}^{2}+k_{y}^{2}\right)\right)$ is the decoupled conduction and valence band quasi-particle energy, *M* is the energy gap, *B* is proportional to the inverse of the effective mass, and we take ħ=1.

The effective parameters *M* and *B* can be derived from a three-dimensional eight-band $\vec{k}\cdot \vec{p}$ theory for a given thickness of HgTe layer [36]. The spin-down Hamiltonian in Equation (Equation 1) is given by $H_{↓}\left(k\right)=H_{↑}^{🟉}\left(-k\right)$.

The Hamiltonian matrix in Equation (Equation 2) can be written in the following compact Weyl-Hamiltonian form (for spin-up):(3)$$H_{↑}=\vec{d}\left(\vec{k}\right)\cdot \vec{\sigma },$$
where pseudospin $\vec{\sigma }$ are the Pauli matrices in the space of electron and heavy hole states, and the vector $\vec{d}\left(\vec{k}\right)$ is given by
(4)$$\vec{d}\left(\vec{k}\right)=\left(v_{f}k_{x},v_{f}k_{y},\frac{\Delta (k_{x},k_{y})}{2}\right).$$

We will show later that the vector $\vec{d}\left(\vec{k}\right)$ maps the in-plane wavevectors $k_{x}$, $k_{y}$ onto a Bloch sphere and characterizes the topology of the bulk energy bands.

### 2.1. Influence of Strain

In this paper, we study the influence of compressive strain in HgTe TI quantum dots as shown in Figure 1b. We account for the effects of strain by introducing a four-band Bir–Pikus Hamiltonian derived from the eight-band Bir–Pikus Hamiltonian [36,42]:(5)$$H_{↑↓}^{BP}\left(k\right)=\left[\begin{array}{cc} \hat{t} & 0\\ 0 & \hat{u}+\hat{v} \end{array}\right],$$
where the operators $\hat{t}=a_{c}\left(\varepsilon_{xx}+\varepsilon_{yy}\right)$, $\hat{u}=a_{v}\left(\varepsilon_{xx}+\varepsilon_{yy}\right)$ and $\hat{v}=\frac{1}{2}b\left(\varepsilon_{xx}+\varepsilon_{yy}\right)$ are written in terms of the strain tensor matrix elements $\varepsilon_{ij}$, and deformation potentials $a_{c}$, $a_{v}$, and *b*. In this paper, we utilize the deformation potential parameters found in van de Walle’s work [43], namely $a_{c}=-4.60$ eV, $a_{v}=-0.13$ eV, and b=−1.15 eV.

Following Novik et al.’s [42] approach, we add the spin-up Bir–Pikus Hamiltonian, Equation (Equation 5), to the spin-up BHZ Hamiltonian, Equation (Equation 2), resulting in the strain Hamiltonian:(6)$$H_{↑}^{Strain}\left(k\right)=\left[\begin{array}{cc} \frac{\Delta (k_{x},k_{y})}{2}-\frac{\gamma (\varepsilon )}{2} & v_{f}\left(k_{x}-ik_{y}\right)\\ v_{f}\left(k_{x}+ik_{y}\right) & -\frac{\Delta (k_{x},k_{y})}{2}+\frac{\gamma (\varepsilon )}{2} \end{array}\right],$$
where $\gamma \left(\varepsilon \right)=\hat{u}+\hat{v}-\hat{t}$ describes the shift of the conduction and valence bands of the HgTe TI [2]. γ(ε) renormalizes the gap such that the Fermi level remains at E=0 and can be found by taking a combination of energies at the bottom of the conduction band $E_{c}\left(k=0\right)$ and top of the valence band $E_{v}\left(k=0\right)$ at the Γ-point: $\left(E_{c}\left(k=0\right)+E_{v}\left(k=0\right)\right)/2=\left(\hat{t}+\hat{u}+\hat{v}\right)/2$ and adding it to the Bir–Pikus Hamiltonian. Here γ(ε) acts as a tuning parameter, driving the TI square quantum dot from the non-trivial, topological phase to the trivial, normal phase. The strain tensors adjust the width and height of the square by $\varepsilon_{xx}=\Delta x/x$ and $\varepsilon_{yy}=\Delta y/y$.

In this work, we consider the case when a square quantum dot is compressively strained in the horizontal and vertical directions as shown in Figure 1b and compare our results to straining 3D HgTe TI disc quantum dots [36]. We then discuss the application of strained quantum dots as quantum strain-based sensors.

### 2.2. Energy Levels and Wavefunctions of HgTe Nanostructures

In finite HgTe nanostructures, the motion of an electron is laterally confined by an external potential V(x,y). The wavefunction $|\varphi_{s}↑\rangle$ of the spin-up electron can be expressed as a linear combination of electron |e↑> and heavy hole |hh↑〉 basis states and envelope functions f(x,y) and g(x,y):(7)$$|\varphi_{s}↑\rangle =A_{s}f\left(x,y\right)|e↑\rangle +B_{s}g\left(x,y\right)|hh↑\rangle .$$

The envelope functions for the quantum dot level *s* satisfy the effective Schrödinger equation: (8)$$\left[\begin{array}{cc} \frac{\Delta ({\hat{p}}_{x},{\hat{p}}_{y})}{2}+V\left(x,y\right) & v_{f}\left({\hat{p}}_{x}-i{\hat{p}}_{y}\right)\\ v_{f}\left({\hat{p}}_{x}+i{\hat{p}}_{y}\right) & -\frac{\Delta ({\hat{p}}_{x},{\hat{p}}_{y})}{2}-V\left(x,y\right) \end{array}\right]\left[\begin{array}{c} A_{s}f\left(x,y\right)\\ B_{s}g\left(x,y\right) \end{array}\right]=E_{s}\left[\begin{array}{c} A_{s}f\left(x,y\right)\\ B_{s}g\left(x,y\right) \end{array}\right],$$
where ${\hat{p}}_{x}=-i\partial /\partial x$ and ${\hat{p}}_{y}=-i\partial /\partial y$ are momentum operators acting on the envelope functions *f* and *g*.

### 2.3. Energy Levels and Wavefunction of HgTe Quantum Disc

Following our earlier work [36], we start our discussion of quantum dots with a circular quantum disc with radius *R* as shown in Figure 1c, where the potential is infinite outside of the disc and zero inside. The wavefunction is a spinor characterized by pairs of angular momentum quantum numbers *m* in the conduction band and m+1 in the valence band for a state *p*. We expand the wavefunction in the basis of Bessel functions as:(9)$$|\Psi_{m}^{p}\left(r\right)\rangle =\sum_{n}A_{n}^{p,m}\phi_{n,m}\left(r\right)|e\rangle +\sum_{s}B_{s}^{p,m+1}\phi_{s,m+1}\left(r\right)|hh\rangle ,$$
where $\phi_{n,m}\left(r\right)=\frac{\sqrt{2}}{R}\frac{1}{|J_{m+1}\left(\alpha_{m}^{n}\right)|}J_{m}\left(\alpha_{m}^{n}\frac{r}{R}\right)\frac{1}{\sqrt{2\pi }}e^{im\varphi }$, $J_{m}\left(\alpha_{m}^{n}\frac{r}{R}\right)$ is the Bessel function of order *m*, and $\alpha_{m}^{n}$ is the *n*-th zero of the Bessel function of order *m*.

The probability of an electron positioned at $r=r_{0}$ is given by:(10)$$\begin{array}{c} \langle \Psi_{m}^{p}\left(r\right)|\delta \left(r-r_{0}\right)|\Psi_{m}^{p}\left(r\right)\rangle =\sum_{n,q}{\left(A_{n}^{p,m}\right)}^{🟉}A_{q}^{p,m}\phi_{n,m}\left(r_{0}\right)\phi_{q,m}\left(r_{0}\right)\\ \;\;+\sum_{s,w}{\left(B_{s}^{p,m+1}\right)}^{🟉}B_{w}^{p,m+1}\phi_{s,m+1}\left(r_{0}\right)\phi_{w,m+1}\left(r_{0}\right). \end{array}$$

The Hamiltonian, neglecting strain, in polar coordinates is given by
(11)$$H_{↑}=\left[\begin{array}{cc} \frac{\Delta ({\hat{k}}_{r})}{2} & v_{f}{\hat{k}}_{-}\\ v_{f}{\hat{k}}_{+} & -\frac{\Delta ({\hat{k}}_{r})}{2} \end{array}\right],$$
where $\frac{\Delta ({\hat{k}}_{r})}{2}=M+B\left(-\frac{\partial^{2}}{\partial r^{2}}-\frac{1}{r}\frac{\partial }{\partial r}-\frac{1}{r^{2}}\frac{\partial^{2}}{\partial \varphi^{2}}\right)$ is an intraband operator, and ${\hat{k}}_{\mp }=-ie^{i\varphi }\left(\frac{\partial }{\partial r}\mp i\frac{1}{r}\frac{\partial }{\partial \varphi }\right)$ are operators connecting conduction and valence band states. Acting with the Hamiltonian in Equation (Equation 11) on the wavefunction given in Equation (Equation 9), we arrive at the set of equations for amplitudes *A* and *B*:(12)$$\varepsilon_{n,m}A_{n}^{m,p}+iv_{f}\sum_{s}\langle m,n|k_{-}|m+1,s\rangle B_{s}^{m+1,p}=E^{m,p}A_{n}^{m,p}$$
(13)$$-iv_{f}\sum_{s}\langle m+1,n|k_{+}|m,s\rangle A_{s}^{m,p}-\varepsilon_{n,m+1}B_{n}^{m+1,p}=E^{m,p}B_{n}^{m+1,p},$$
where $\varepsilon_{n,m}=\left(M+B{\left(\alpha_{m}^{n}\right)}^{2}\right)$ and
(14)$$\langle m,n|k_{-}|m+1,s\rangle =\frac{2}{R}\frac{J_{m+1}\left(\alpha_{m}^{n}\right)}{|J_{m+1}\left(\alpha_{m}^{n}\right)|}\frac{J_{m+2}\left(\alpha_{m+1}^{s}\right)}{|J_{m+2}\left(\alpha_{m+1}^{s}\right)|}\frac{\alpha_{m}^{n}\alpha_{m+1}^{s}}{{\left(\alpha_{m}^{n}\right)}^{2}-{\left(\alpha_{m+1}^{s}\right)}^{2}},$$
(15)$$\langle m+1,n|k_{+}|m,s\rangle =\frac{2}{R}\frac{J_{m+1}\left(\alpha_{m+1}^{n}\right)}{|J_{m+1}\left(\alpha_{m+1}^{n}\right)|}\frac{J_{m+2}\left(\alpha_{m}^{s}\right)}{|J_{m+2}\left(\alpha_{m}^{s}\right)|}\frac{\alpha_{m+1}^{n}\alpha_{m}^{s}}{{\left(\alpha_{m+1}^{n}\right)}^{2}-{\left(\alpha_{m}^{s}\right)}^{2}}.$$

The eigenvectors $A_{n}^{m,p}$ and $B_{s}^{m,p}$, and eigenvalues $E^{m,p}$ are found by diagonalizing the Hamiltonian in Equations (Equation 12) and (Equation 13). The spin-down eigenvalues can be obtained analogously.

### 2.4. Energy Levels and Wavefunction of HgTe Quantum Square

Let us now consider an HgTe square TI quantum dot, finite in the *x*- and *y*-directions with side length *a* as shown in Figure 1a. We seek the wavefunction for an electron in a square quantum well given in terms of trigonometric functions $f_{n,m}\left(x,y\right)$, where *n* and *m* are integer quantum numbers. Our wavefunction has the form:(16)$$|\Psi^{p}\left(x,y\right)\rangle =\sum_{n,m}A_{n,m}^{p}f_{n,m}\left(x,y\right)|e\rangle +\sum_{k,l}B_{k,l}^{p}f_{k,l}\left(x,y\right)|hh\rangle ,$$
where $f_{n,m}\left(x,y\right)=\frac{2}{a}sin\left(\frac{n\pi x}{a}\right)sin\left(\frac{m\pi y}{a}\right)$. The function $f_{n,m}\left(x,y\right)$ vanishes at the edges of the square.

Here the probability density of an electron positioned at $\vec{r}=\vec{r_{0}}$ can be found by:(17)$$\begin{array}{c} \langle \Psi^{p}\left(x,y\right)|\delta (\vec{r}-\vec{r_{0}})|\Psi^{p}\left(x,y\right)\rangle =\\ \sum_{n,m,q,r}{\left(A_{n,m}^{p}\right)}^{🟉}A_{q,r}^{p}f_{n,m}\left(x_{0},y_{0}\right)f_{q,r}\left(x_{0},y_{0}\right)+\sum_{k,l,v,w}{\left(B_{k,l}^{p}\right)}^{🟉}B_{v,w}^{p}f_{k,l}\left(x_{0},y_{0}\right)f_{v,w}\left(x_{0},y_{0}\right). \end{array}$$

Acting with the Hamiltonian, Equation (Equation 6), on the wavefunction, Equation (Equation 16), gives us a system of equations for coefficients $A_{q,r}^{p}$ and $B_{q,r}^{p}$:(18)$$\left[M+B\left(\frac{q^{2}\pi^{2}}{a^{2}}+\frac{r^{2}\pi^{2}}{a^{2}}\right)-\frac{\gamma (\varepsilon )}{2}\right]A_{q,r}^{p}+v_{f}\sum_{kl}\langle qr|V|kl\rangle B_{kl}^{p}=E^{p}A_{q,r}^{p},$$
(19)$$+v_{f}\sum_{nm}\langle qr|V|nm\rangle A_{nm}^{p}-\left[M+B\left(\frac{q^{2}\pi^{2}}{a^{2}}+\frac{r^{2}\pi^{2}}{a^{2}}\right)-\frac{\gamma (\varepsilon )}{2}\right]B_{q,r}^{p}=E^{p}B_{q,r}^{p},$$
with strain given by parameter γ(ε) and coupling matrix elements given by $\langle qr|V|kl\rangle =-i\delta_{r,l}\langle q|\partial_{x}|k\rangle -\delta_{q,k}\langle r|\partial_{y}|l\rangle$, and $\langle q|\partial_{x}|k\rangle =\langle q|\partial_{y}|k\rangle =\frac{2}{a}\frac{qk}{\left(k^{2}-q^{2}\right)}\left[{\left(-1\right)}^{q+k}-1\right]$ if q≠k otherwise $\langle q|\partial_{x}|k\rangle =\langle q|\partial_{y}|k\rangle =0$ when q+k is even and when q=k. The spin-up eigenvectors and eigenvalues can be found by diagonalizing Equations (Equation 18) and (Equation 19) while the spin-down eigenvectors and eigenvalues can be found similarly.

## 3. Results and Discussion

We start our discussion with a study of the transition of the BHZ Hamiltonian from a topological phase to a normal phase as a function of its parameters. This is conducted by mapping the $\left(k_{x},k_{y}\right)$ plane onto the normal vector $\hat{n}\left(k_{x},k_{y}\right)=\vec{d}\left(k_{x},k_{y}\right)/\left|\vec{d}\left(k_{x},k_{y}\right)\right|$ defined on a Bloch sphere following the analysis for topological insulators and superconductors as discussed by, e.g., Alicea [44]. We follow with an analysis of the electronic properties of disc HgTe TI quantum dots and square HgTe TI quantum dots with and without applied strain obtained from numerically diagonalizing their respective Hamiltonians discussed in Section 2. We then relate the emergence of edge states to strain by tuning the parameter γ(ϵ) until a topological phase transition occurs in square quantum dots.

### 3.1. Phase Transitions in the Bulk BHZ Model

Here we discuss the existence of two phases in the BHZ Hamiltonian: the trivial (normal) and the non-trivial (topological). The trivial insulator leads to normal energy bands with |hh〉 states contributing to the valence band and |e〉 states contributing to the conduction band, while the non-trivial phase leads to a TI with inverted bands and edge states existing inside the bulk energy gap.

The bulk BHZ Hamiltonian, Equation (Equation 3), $H_{↑}=\vec{d}\left(k_{x},k_{y}\right)\cdot \vec{\sigma }$, is entirely specified by the vector $\vec{d}\left(k_{x},k_{y}\right)$. We relate the Hamiltonian to the topology by defining a vector $\hat{n}\left(k_{x},k_{y}\right)=\vec{d}\left(k_{x},k_{y}\right)/\left|\vec{d}\left(k_{x},k_{y}\right)\right|$. The vector $\hat{n}$ maps the $\left(k_{x},k_{y}\right)$ plane onto the Bloch sphere. The topology of the mapping depends on parameters *M*, *B*, and $v_{f}$.

Figure 2a shows the values of $\hat{n}\left(k_{x},k_{y}\right)$ mapped onto a Bloch sphere for M<0 (inverted bands) and B>0. We see that the mapping converts the $\left(k_{x},k_{y}\right)$ plane onto the entire Bloch sphere. In Figure 2b, we show the result of the mapping for M>0 (normal bands) and B>0, i.e., the sign of the energy gap is reversed. For this set of parameters, only the top part of the sphere is populated and there is a hole on the Bloch sphere. Clearly, the topology of the Bloch sphere is different for the trivial (M>0) and topological (M<0) insulators. Therefore, we can create a trivial or topological insulator by changing the sign of *M* and inverting the bands.

### 3.2. Edge States in the Disc Quantum Dot

We now turn to the disc quantum dot. By comparison of results obtained using a quasi-2D BHZ model with 3D eight band $\vec{k}\cdot \vec{p}$ model we aim to validate the BHZ model and develop better understanding of results obtained in the eight band model. In a disc the angular momentum ${\hat{L}}_{z}=m+{\hat{s}}_{z}$ is conserved. Here, ${\hat{s}}_{z}=\pm 1/2$ is the spin of the electron (up or down). For ${\hat{s}}_{z}=1/2$, the energy spectrum as a function of ${\hat{L}}_{z}$ is obtained via a numerical diagonalization of Equations (Equation 12) and (Equation 13). For the opposite electron spin we formulate the appropriate equations arising from the Hamiltonian $H_{↓}$. The eigenstates obtained for ${\hat{s}}_{z}=1/2$ (${\hat{s}}_{z}=-1/2$) are shown in Figure 3a with black (red) bars.

Figure 3a shows the energy levels of a disc with radius R=167 nm for parameters corresponding to the topological, inverted band regime with M=−150 meV. We see a discrete spectrum of valence and conduction band states. Additionally, inside the energy gap, from −150 meV to 150 meV, we find a ladder of equally spaced edge states with linear dispersion as a function of angular momentum and energy bands. We see that the disc behaves like a finite edge of an HgTe TI nanoribbon [10], with periodic boundary conditions yielding a discrete energy spectrum with size quantization related to the circumference of the disc.

Figure 3b shows the electronic probability as a function of the radial coordinate for different edge states labeled in Figure 3a. This electronic probability density is calculated using Equation (Equation 10), by summing the eigenvectors for the angular momenta states in Figure 3a. It is found that the electronic probability density peaks away from the physical edge and decays quickly into the center of the disc.

These results show that the electronic properties of the 2D four-band BHZ HgTe TI disc quantum dot are in agreement with previous works on 3D eight-band HgTe TI disc quantum dots by some of us [36] and agree with BHZ results by Chang et al. [31]. Thus, we anticipate that the electronic properties and topological phase transitions found when compressively straining the 3D eight-band HgTe TI quantum dots will also be present when compressively straining the 2D four-band BHZ HgTe TI quantum dots.

### 3.3. Edge States and Strain-Driven Transitions in the Square Quantum Dot

With validation of the BHZ model we now turn to discussion of the energy spectrum of a square quantum dot as a function of strain obtained by diagonalizing Equations (Equation 18) and (Equation 19). The main difference between the quantum disc and a quantum square is the presence of sharp corners. We show below that these corners have zero electronic probability density, with edge states localized along the edges of the square. Figure 4a–d shows the evolution of the energy spectrum of the square quantum dot with lateral size a=40 nm and inverted bands as a function of applied strain while Figure 4e–h shows probability density of the corresponding lowest energy conduction band state. We see a discrete spectrum of quantized energy levels in the conduction and valence bands. In the topological inverted band regime (zero strain) in the energy window corresponding to a bulk gap, we see a ladder of equally spaced energy levels, Figure 4a, with corresponding probability density localized at the edges of the square and expelled from the corners Figure 4e. In Figure 4a–d, only the bulk bands are plotted as a function of momentum.

In Figure 4e, the electronic probability density is plotted for the first energy state above the Fermi level in Figure 4a. The electronic probability density is obtained using Equation (Equation 17). Figure 4e also shows the formation of edge states localized along the edges of the square but approaching zero in the corners of the square. These edge states can be viewed as quantum rings [31], where electrons are localized along the sides of the squares and can tunnel from site to site. Thus, these edge states may be used as sites in the 1D Hubbard model.

Figure 4 shows that by progressively applying compressive stress in the horizontal and vertical directions of the square quantum dot one causes the inverted bands to close. Then, as depicted in Figure 4c,g, at the topological phase transition $\varepsilon_{xx}=-0.0385$ and $\varepsilon_{yy}=-0.0385$ , the bands form a Dirac cone. Upon applying compressive strain beyond this threshold, the bands become normally ordered, a gap free of edge states opens and states are localized inside the quantum dot. Therefore, under negative strain, the system is driven from the inverted band regime to the normal band regime and edge states disappear. Presence or absence of edge states can be detected in transport through the quantum dot. This is the principle of operation of our quantum strain sensor [36] based on TI HgTe quantum dots.

## 4. Conclusions

We present here a contribution to the theory of the electronic properties of quasi two-dimensional nanostructures made of topological insulators (TIs). The TI quantum dots were described by the four-band Bernevig–Hughes–Zhang (BHZ) Hamiltonian as well as a 3D eight band $\vec{k}\cdot \vec{p}$ model [36]. The trivial versus topological properties of the BHZ Hamiltonian were inferred from the mapping of the 2D wavevector plane through the BHZ Hamiltonian onto a Bloch sphere. In the topologically non-trivial case, edge states were found in the disc and square geometries. By tuning the compressive strain on the square quantum dot in the topologically non-trivial phase, the edge states began to disappear from the energy gap and after a significant amount of strain was applied, a topological phase transition occurred, causing the transition of TI to normal insulator and vanishing of edge states in a quantum dot. This allowed us to relate the emergence of edge states explicitly to transition from the normal to the inverted band regime tuned by applied strain to the square quantum dot and to the change in the topology of mapping the BHZ Hamiltonian onto Bloch spheres.

We show that for a square quantum dot edge states have vanishing probability density at the corners so electrons are localised along each edges. These localised states would play the role of sites in the Hubbard model once they are populated with electrons which would interact strongly if they are on the same site. The existence of localized 1D edge states may enable the design of quasi-one-dimensional quantum rings with localized electronic states along the sides of the square, acting as tunable one-dimensional Hubbard models once populated with interacting electrons. The presence or absence of edge states and hence modification of their electronic properties is found to be controlled by strain. Strain-driven topological phase transitions can be detected in transport and serve as a basis for quantum strain sensors based on HgTe quantum dots in topological insulators.

## Figures and Tables

**Figure 1 nanomaterials-12-04283-f001:**
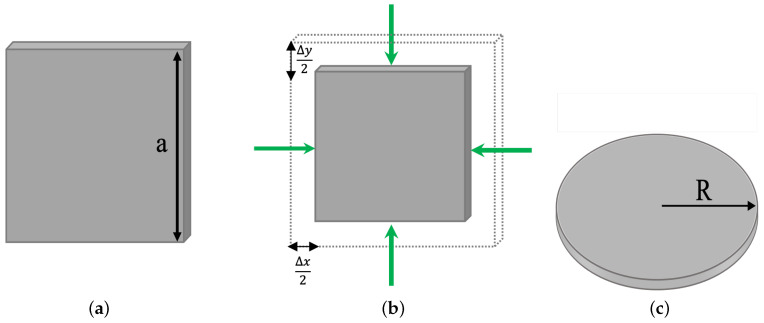
Schematic pictures of quasi-2D HgTe TI quantum dot systems. (**a**) Square quantum dot unstrained. (**b**) Square quantum dot after undergoing compressive strain in the horizontal and vertical directions resulting in the quantum dot reducing in size in both directions. (**c**) Disc quantum dot.

**Figure 2 nanomaterials-12-04283-f002:**
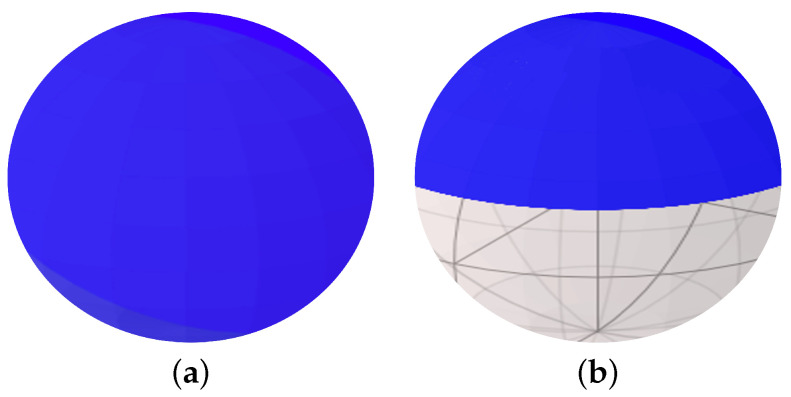
Mapping $\vec{n}$ of the $\left(k_{x},k_{y}\right)$ plane represented by $-10<k_{x}<10$ and $-10<k_{y}<10$ region (with wavectors in nm${}^{-1}$) onto a Bloch sphere for the (**a**) topological, inverted band regime where M<0 and B>0, and (**b**) topologically trivial regime with normally ordered bands for M>0 and B>0. The topology of the two Bloch spheres is different, (**b**) contains a hole and (**a**) does not. The parameters of the topological regime are M=−150 meV, B=107 meVnm${}^{2}$, and $v_{f}=600$ meVnm, while in the topologically trivial regime the parameters are M=+150 meV, B=107 meVnm${}^{2}$, and $v_{f}=600$ meVnm [36].

**Figure 3 nanomaterials-12-04283-f003:**
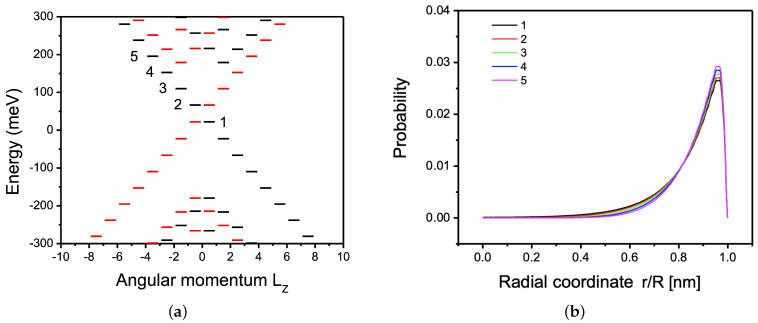
(**a**) Spectrum of energy levels for a disc TI quantum dot as a function of total angular momentum $L_{z}=m+S_{z}$ where *m* is the orbital angular momentum and $S_{z}$ is the spin for radius R=167 nm. Red levels correspond to spin down and black levels to spin up. Energy levels in the bulk gap are visible. (**b**) Electronic probability for states 1–5 shown in (**a**). The edge character of gap states and their position with respect to the edge are visible. The Hamiltonian in Equations (Equation 12) and (Equation 13) is numerically diagonalized with parameters M=−150 meV, B=107 meVnm${}^{2}$, and $v_{f}=600$ meVnm, corresponding to the topological regime.

**Figure 4 nanomaterials-12-04283-f004:**
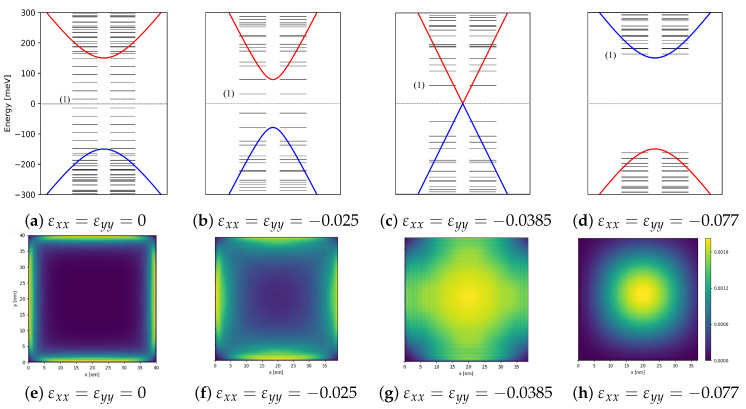
Energy spectra for four different cases of applied strain. (**a**) No strain applied, a topological insulator with a ladder of equally spaced energy levels in the gap of the bulk material is visible. (**b**) Some strain applied, the gap begins to close. (**c**) Dirac cone without edge states. (**d**) Normal insulator without states inside of the bulk gap. We see that by compressively straining the system, a topological phase transition occurs. The first energy level is labelled (1) and spin degenerate eigenvalues are shown next to each other and plotted together with the energy bands to show edge states inside the energy gap. The black dotted line at zero energy denotes the Fermi level. (**e**–**h**) Greyscale showing the probability of the first state above the Fermi level in (**a**–**d**). The parameters used in the diagonalization of the square quantum dot are the topological regime parameters: M=−150 meV, B=107 meVnm${}^{2}$, and $v_{f}=600$ meVnm.

## Data Availability

Data supporting results are available from the corresponding author upon request.

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
