# Peer review of "Edge States and Strain-Driven Topological Phase Transitions in Quantum Dots in Topological Insulators"

_nanomaterials, 2022, doi:10.3390/nano12234283_

Round 1

Reviewer 1 Report

The paper is devoted to the study of the electronic properties of quasi two-dimensional quantum dots made of topological insulators. Specifically, the topological insulator is described by Bernevig-Hughes-Zhang Hamiltonian, supplemented by the Bir-Pikus Hamiltonian. In this way, the Authors take into account the effects of compressive strain in topological insulator quantum dots.

Among the results, it is shown that the trivial versus topological properties of the model Hamiltonian are characterized by different topologies, related to the mapping of the in-plane wavevectors through onto a Bloch sphere.  Moreover, in the topologically nontrivial case, edge states are formed when circular and square geometries of the quantum dot are concerned. If the strain is implemented, topological phase transitions between topological and trivial phases are produced, resulting in vanishing edge states.  Interestingly, this latter conclusion may have potential application, i. e. it is useful to design quantum strain sensors based on strain-driven transitions in HgTe topological insulator square quantum dots.

The paper is well written and deserves publication. Nevertheless, before publication I suggest to the Authors to comment how the presence or absence of edge states can be detected in transport measurements through the quantum dot. Specifically, identifying the experimental probes that can be used to this scope.

Author Response

Response to Referee’s comments including all changes made.

Referee 1

We thank the referee 1 for his positive comments. Here we respond to his questions and list all changes made to the manuscript.

Referee 1, Question 1: How presence or absence of edge states can be detected in transport measurements through the quantum dot. Specifically, identifying the experimental probes that can be used to this scope.

Our response: The presence or absence of conducting edge states drastically changes the conductivity of the quantum dot once it is connected to source and drain. Hence one can detect strain induced trivial to topological transition from current through a quantum dot. This has been described in detail in our previous publication, Ref. 36. As this is not obvious, we added a sentence on page 1 connecting presence and absence of edge states to quantum dot conductivity.

Manuscript changes: On page 2 we added:

When our quantum dot is connected to the leads, the conductivity strongly depends on the presence or absence of edge states induced by strain. Hence strain in the quantum dot can be detected through conductance measurements in this nano strain sensor [36].

Reviewer 2 Report

The paper theoretically addresses the effect of compressive strain on the electronic and topological properties of 2D quantum dots with square shape. As a model they use the four-band BHZ Hamiltonian. In particular, they consider the energy spectrum and probability distribution of the edge states. They found that by increasing the strain it is possible to induce the transition from the topological to the trivial state.

The paper is a straightforward extension of Ref. [36] by the same authors.

The paper deserves publication in Nanomaterials only when the following comments are properly addressed and the paper changed accordingly:

1) Regarding the discussion of Fig. 2b, the meaning of the wording "the Bloch sphere contains a hole" is not clear to me. If I understand correctly the figure, the vector d does not cover entirely the Bloch sphere by spanning a range of values of kx and ky. Is that correct? Which is the range considered? Is the sphere only half covered or a different portion is covered? Does this depend on the actual values of the parameters (once M>0)? The authors could spell out these issues and find a different wording.

2) The mapping between wavevector and vector d is never used in the rest of the paper and I do not understand the reason to even introduce it. Can it be used in the presence of strain? The authors should address this point otherwise the mapping can be removed from the paper.

3) Regarding Fig. 2b, the Bloch sphere is completely covered for any choice of parameters, as long as M<0? The authors should comment on that if the figure will not be removed.

4) Regarding Fig. 3a, what is the difference between levels in red and in black? Moreover, I think it is improper to refer to the spectrum of edge states as "Dirac cone", since it is a function of the eigenvalues of Lz and not of momentum. Is Lz the z component of the total angular momentum? Please specify.

5) Regarding Fig. 4a-d, for a better understanding of the figure, I would put the momentum axis of the bulk energy bands. Moreover, in the caption it should be specified explicitly that the energy bands refer to the bulk.

6) Finally, I do not understand the sentence "... edge states may be used as sites in the 1D Hubbard model." A edge state is basically a conducting non-interacting channel. How can this be seen as a site? Where is the on-site interaction?

Author Response

Response to Referee’s comments including all changes made.

Referee 2.

We thank Referee 2 for careful reading of the manuscript and several questions aiming at improving the readability of our manuscript.

Referee 2 - Question 1:

Regarding the discussion of Fig. 2b, the meaning of the wording "the Bloch sphere contains a hole" is not clear to me. If I understand correctly the figure, the vector d does not cover entirely the Bloch sphere by spanning a range of values of kx and ky. Is that correct? Which is the range considered? Is the sphere only half covered or a different portion is covered? Does this depend on the actual values of the parameters (once M>0)? The authors could spell out these issues and find a different wording.

Our response: We thank the referee for his questions. We note that the mapping of parameter space, here kx and ky, onto Bloch sphere using vector d/|d| is a standard procedure in determining topologically different properties of our Hamiltonian. This is described in a number of reviews on topological insulators and superconductors. We added here reference to Jason Alicea Rep. Prog. Phys. 75 (2012) publication. For Fig2, we take a grid of k values in the ranges:  and   ( k in nm_1) and map (kx,ky)  through d(k)/|d(k)| onto Bloch sphere.  The topology of the Hamiltonian is now reflected in the topology of the coverage of the Bloch sphere. We have added information about the range of kx,ky in the figure 2 caption. In the trivial regime corresponding to M>0, only fraction of the Bloch sphere is covered while for M<0 entire Bloch sphere is covered. The fractional coverage can be understood in terms of hole present or absent in fully covered sphere hence the topology of Hamiltonian is different for M>0 vs M<0. This result is essential in the remainder of the paper.

Referee 2 - Question 2:

The mapping between wavevector and vector d is never used in the rest of the paper, and I do not understand the reason to even introduce it. Can it be used in the presence of strain? The authors should address this point otherwise the mapping can be removed from the paper.

Our response:  We note again that the mapping of parameter space, here kx and ky, onto Bloch sphere of vector d/|d| is a standard procedure in determining topologically different properties of our Hamiltonian. This is described in a number of reviews, we added here reference to Jason Alicea, Rep. Prog. Phys. 75 (2012). The mapping connects electronic structure calculations with topology. The mapping can be used in the presence of strain which renormalizes M.

Manuscript changes: On page 6, below section Results and Discussion, we added a sentence and reference to Alicea:

We start our discussion with a study of the transition of the BHZ Hamiltonian from a topological phase to a normal phase as a function of its parameters. This is done by mapping the {$(k_x,k_y)$} plane onto the normal vector  {$\hat{n}(k_x,k_y)=\vec{d}(k_x,k_y)/|\vec{d}(k_x,k_y)|$} defined  on a Bloch sphere following the analysis for topological insulators and superconductors as discussed by e.g. Alicea\cite{alicea2012}.:

Referee 2 - Question 3:

Regarding Fig. 2b, the Bloch sphere is completely covered for any choice of parameters, as long   as M<0? The authors should comment on that if the figure will not be removed.

Our response: Note: Fig. 2b is the case where M>0, Fig 2a corresponds to M<0 and completely covered Bloch sphere. The Bloch sphere is completely covered irrespective of details of parameters which is why topological effects are robust.

Referee 2 Question 4:

Regarding Fig. 3a, what is the difference between levels in red and in black? Moreover, I think it is improper to refer to the spectrum of edge states as "Dirac cone", since it is a function of the eigenvalues of Lz and not of momentum. Is Lz the z component of the total angular momentum? Please specify.

Our response:  In Fig.3 red levels correspond to spin down and black levels to spin up. We added this information to Fig.3 caption, and we thank the Referee for pointing out this omission. The total angular momentum Lz=m+sz is a sum of orbital m and spin sz angular momenta.

Manuscript changes: 

In the second paragraph of page 7, we added the sentence: In a disc the angular momentum Lz = m + sz is conserved. Here, sz = ±1 / 2 is the spin of the electron (up or down). For sz = 1 / 2, the energy spectrum as a function of Lz is obtained via a numerical diagonalization of Eqs. (12) and (13). For the opposite electron spin we formulate the appropriate equations arising from the Hamiltonian H↓. The eigenstates obtained for sz = 1/2 (sz = −1/2) are shown in FIG. 3a with black (red) bars.

In caption of Fig.3, we added this sentence:

Spectrum of energy levels for a disc TI quantum dot as a function of total angular momentum {$L_z=m+s_z$} where $m$ is the orbital angular momentum and $s_z$ is the spin for radius {$R=167$}nm. Red levels correspond to spin down and black levels to spin up states.

Finally, following the Referee’s suggestion, we have removed the remark about the Dirac cone and replaced it with “linear energy dispersion”.

 Referee 2 -Question 5:

Regarding Fig. 4a-d, for a better understanding of the figure, I would put the momentum axis of the bulk energy bands. Moreover, in the caption it should be specified explicitly that the energy bands refer to the bulk.

Our response: We thank the referee for his suggestion. However, a momentum axis may confuse readers as only the bulk bands depend on the momentum while the eigenvalues in a quantum dot do not depend on the momentum, so the reader may get confused to think that the eigenvalue is somehow related to the momentum here. We propose to retain current figure, but we added a statement on page 8, section 3.3, paragraph 1: explaining that “In FIG. 4a-4d, only the bulk bands are plotted as a function of momentum.  

Referee 2 - Question 6:

 Finally, I do not understand the sentence "... edge states may be used as sites in the 1D Hubbard model." A edge state is basically a conducting non-interacting channel. How can this be seen as a site? Where is the on-site interaction?

Our response and manuscript changes: We added this sentence on page 9 in section “Conclusions”. 

We show that for a square quantum dot edge states have vanishing probability density at the corners, so electrons are localised along each edge. These localised states would play the role of sites in the Hubbard model once they are populated with electrons which would interact strongly if they are on the same site. The existence of localized 1D edge states may enable the design of quasi-one-dimensional quantum rings with localized electronic states along the sides of the square, acting as tunable one-dimensional Hubbard models once populated with interacting electrons.